# Stratospheric water vapor affecting atmospheric circulation

Edward Charlesworth [1,17] ✉, Felix Plöger[1,2,17], Thomas Birner[3], Rasul Baikhadzhaev [1], Marta Abalos[4], Nathan Luke Abraham [5,6], Hideharu Akiyoshi [7], Slimane Bekki [8], Fraser Dennison[9], Patrick Jöckel [10], James Keeble[5,6], Doug Kinnison[11], Olaf Morgenstern [12], David Plummer [13], Eugene Rozanov [14], Sarah Strode [15,16], Guang Zeng [12], Tatiana Egorova[14] & Martin Riese [1]

Water vapor plays an important role in many aspects of the climate system, by affecting radiation, cloud formation, atmospheric chemistry and dynamics. Even the low stratospheric water vapor content provides an important climate feedback, but current climate models show a substantial moist bias in the lowermost stratosphere. Here we report crucial sensitivity of the atmospheric circulation in the stratosphere and troposphere to the abundance of water vapor in the lowermost stratosphere. We show from a mechanistic climate model experiment and inter-model variability that lowermost stratospheric water vapor decreases local temperatures, and thereby causes an upward and poleward shift of subtropical jets, a strengthening of the stratospheric circulation, a poleward shift of the tropospheric eddy-driven jet and regional climate impacts. The mechanistic model experiment in combination with atmospheric observations further shows that the prevailing moist bias in current models is likely caused by the transport scheme, and can be alleviated by employing a less diffusive Lagrangian scheme. The related effects on atmospheric circulation are of similar magnitude as climate change effects. Hence, lowermost stratospheric water vapor exerts a first order effect on atmospheric circulation and improving its representation in models offers promising prospects for future research.

Water vapor represents the most impactful trace gas in the atmosphere, approximately doubling the warming due to carbon dioxide alone through positive climate feedbacks[1], and also plays a first-order role in atmospheric energetics via cloud-radiative effects and latent heat release. In the stratosphere, water vapor concentrations are very low and clouds are essentially absent. Nevertheless, local radiative cooling due to water vapor still plays an important role in controlling the stratospheric temperature structure[2,3], especially just above the tropopause[4]. As the global radiation budget is particularly sensitive to the water vapor content in the upper troposphere and lower

stratosphere, even small water vapor variations in that region can cause large radiative perturbations and climate impacts[5,6].

Climate models predict a robust increase of stratospheric water vapor in response to future climate change, particularly in the extratropical lowermost stratosphere[7]. Indications for such a long-term stratospheric water vapor increase have been found in observations[8], especially after the turn of the millenium[9]. This stratospheric water vapor increase alone induces a positive climate feedback[10], which may cause about 10% of the simulated global mean surface temperature increase, contributing significantly to Arctic amplification, modifying

atmospheric circulation[11], and even affecting the poleward expansion of the Hadley cells[12]. However, the simulated stratospheric water vapor feedback shows a substantial spread between different models (varying between 0.1 and 0.3 W/(m² K)), similar to the model spread in cloud or ice-albedo feedbacks[13].

In the lower stratosphere, water vapor concentrations are primarily governed by temperatures near the cold tropical tropopause, where air ascends in the upward branch of the global stratospheric circulation[14–16]. In addition, methane oxidation acts as an additional chemical source of water vapor in the stratosphere. Just above the extratropical tropopause, high water vapor concentrations are sporadically mixed from the upper tropical troposphere into the lowermost stratosphere[17], causing excessive moisture compared to stratospheric background values.

Severe model biases regarding transport of water vapor across the tropopause and through the stratosphere pose a long-standing challenge to climate modeling[7,18]. In particular, numerical transport schemes employed in current climate models struggle to accurately represent tracer transport in the presence of strong background gradients, such as for water vapor near the tropopause[19]. For example, commonly employed flux-form semi-Lagrangian transport schemes suffer from numerical diffusion, causing excessive water vapor transport into the stratosphere. As a solution to reduce spurious numerically diffusive transport, the application of Lagrangian transport schemes has been proposed[20]. Although applying such flow-following Lagrangian schemes appears to be an elegant and efficient opportunity to improve model transport, this early attempt was not pursued further in modern high-top models.

The water vapor bias in current climate models is particularly large in the lowermost stratosphere[7] where radiation is most sensitive to water vapor changes[6] and causes associated temperature biases. These temperature biases may, in turn, provoke adverse effects on the general circulation which have not been explored much in the past. Hence, differences in water vapor content in the lowermost

stratosphere across different models may be associated with a range of related circulation effects, including the models' response to increasing Greenhouse gas levels. In general, the moist bias in climate models (factor 2-3[7]) induces large uncertainty in simulations and improving the representation of stratospheric water vapor in models offers an exciting opportunity to reduce uncertainty in future projections.

## Results and discussion
### Lowermost stratospheric water vapor in models
The new suite of climate model simulations from CMIP6 and CCMI−2022 shows a general and tremendous moist bias in the extratropical lowermost stratosphere compared to SWOOSH satellite observations (Methods), exceeding 200% for the multi-model mean (MMM) in the summer hemisphere (Fig. 1) and even larger for individual models (Supp. Figs. 1–4). During summer, the subtropical jet is especially weak, and thereby its effectiveness as a barrier against troposphere-to-stratosphere transport is weak as well. The fact that the starkest contrasts occur below the core of the subtropical jet and at its upper flank points to isentropic mixing as a potential cause, which is known to play a crucial role for moistening the lowermost stratosphere[17,21]. In the winter hemisphere lowermost stratosphere, the moist bias in models is weaker, but still reaches 100% in the multi-model mean, and can be significantly larger in single models. The weak dry bias in models in the overworld stratosphere above -100 hPa is likely related to a known cold bias in tropopause temperatures in models[22] and underrepresented methane oxidation, but the radiative effect of this bias is relatively minor compared to the biases in the lowermost stratosphere[6].

We examined the impacts of stratospheric water vapor on atmospheric temperatures and circulation and explored the role of model transport schemes in the model moist bias by carrying out an experiment with the comprehensive climate model EMAC[23], wherein two different transport schemes were applied. The first model version applies the standard Flux-Form Semi-Lagrangian transport scheme[24],

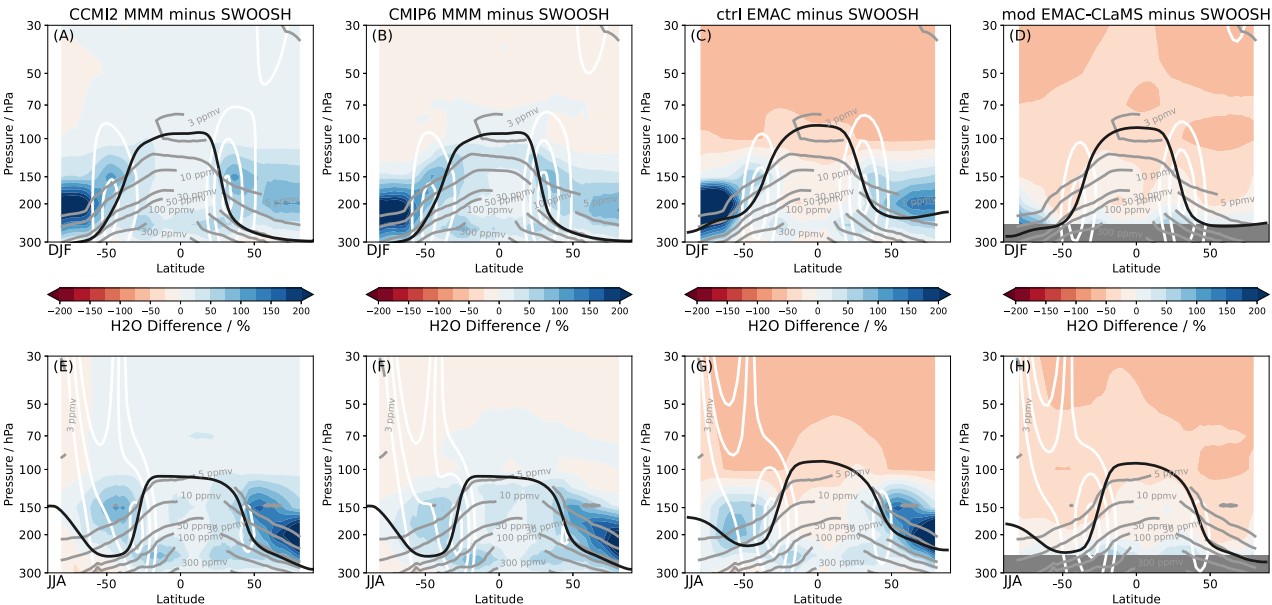

**Fig. 1 | Lowermost stratospheric moist bias in climate model simulations.**
Relative differences in climatological water vapor distributions compared to SWOOSH merged satellite observations for multi model mean from CCMI−2022 (**A**, **E**), CMIP6 (**B**, **F**), control EMAC (**C**, **G**), and Lagrangian modified EMAC−CLaMS simulation (**D**, **H**), for boreal winter (December−February, top) and summer (June−August, bottom). Relative differences are calculated by the relevant distribution minus SWOOSH as a percentage of local SWOOSH values. The SWOOSH water

vapor distribution is shown in light gray solid contours, the 20, 30, and 40 m/s zonal wind contours from the relevant distributions are shown in white, and the WMO lapse rate tropopause as a thick black line. The shown meteorological data is from the respective model simulations, the tropopauses shown for CMIP6 and CCMI2 are from ERA5 reanalysis ("Methods" section). Different periods have been used to calculate the climatologies for CCMI−2022 (2000–2018), CMIP6 (2000–2014), SWOOSH (2000–2014), and EMAC (entire experimental period, "Methods" section).

termed control "ctrl EMAC", and the second a purely Lagrangian scheme (modified "mod" EMAC–CLaMS), which was developed within the Chemical Lagrangian Model of the Stratosphere CLaMS[25] to improve the representation of stratospheric transport[26,27]. In the modified EMAC–CLaMS simulation, only stratospheric water vapor from the Lagrangian calculation is fed into the radiation scheme, to allow isolating radiative effects of stratospheric water vapor ("Methods" section).

The control model simulation produces very similar water vapor distributions when compared to the multi-model means of CMIP6 and CCMI–2022 with high water vapor biases compared to satellite observations in the lowermost stratosphere (Fig. 1), particularly in the summer hemisphere (up to 400%). Compared to the standard model version, the new Lagrangian modified simulation shows substantially reduced lowermost stratospheric water vapor, and very similar values when compared to satellite observations (Fig. 1). These water vapor differences point to reduced moisture transport across the extratropical tropopause in the Lagrangian modified simulation, owing to reduced numerical diffusion compared to the control simulation.

## Effects on atmospheric circulation

The strong sensitivity of simulated lowermost stratospheric water vapor to the choice of the model transport scheme offers an opportunity to investigate the effects of stratospheric water vapor on atmospheric circulation. The model experiment shows that the moist bias causes a significantly colder lowermost stratosphere in control EMAC compared to the new Lagrangian modified EMAC–CLaMS simulation (Fig. 2), due to water vapor induced long-wave cooling. These temperature differences are well above 6 K around the 200 hPa level in the summer hemisphere (reaching peak values of ~10 K, not shown). The cooling due to increased lowermost stratospheric water vapor decreases the equator-to-pole temperature gradient in the subtropics which, in turn, is associated with strengthened zonal winds at the upper flanks of the subtropical jets via thermal wind balance (Methods). As a result, the subtropical jets shift upward and slightly poleward when lowermost stratospheric water vapor is increased (Fig. 2B, F). In the stratosphere, the strengthening of the upper flanks of the subtropical jets induces an upward shift of critical layers for wave breaking, enhancing the wave forcing of the stratospheric circulation (Fig. 2). The wave drag changes caused by stratospheric water vapor changes result in an ~15% stronger tropical upwelling mass flux in the moist-biased control EMAC simulation (Fig. 2C, G).

In the mechanistic model experiment, the stratospheric water vapor induced circulation changes extend downwards into the troposphere (Fig. 2B, F) and cause a complex pattern in near-surface zonal wind response at 850 hPa (Fig. 3a, c). A robust poleward shift of the eddy-driven tropospheric jet in response to increased lowermost stratospheric water vapor is found in the Southern hemisphere during winter (June–August), while the Northern hemisphere signal is similar, but weaker. These poleward jet shifts may partly be related to increased tropopause-level baroclinicity and an upward shift of the tropopause, similar to the general response to global warming[28,29]. However, other effects are at play which are known to force opposite, equatorward jet shifts. Such effects include stratosphere–troposphere coupling of the weakened stratospheric polar vortex due to increased wave driving[30] and a sharpened tropopause[31] for a moister lowermost stratosphere. The former effect is strongest during Northern winter (December–February), while the latter is strongest in the summer hemisphere (Supp. Fig. 5), such that the clearest net poleward shifts emerge in the Southern hemisphere during June–August (Fig. 3).

Although the general zonal mean reponse in the Northern hemisphere is weak, clear regional jet shifts occur particularly in the Atlantic and Pacific sectors (Fig. 3b, d). Although regional effects might be affected by internal variability and are likely model-dependent, our results provide evidence that the representation of water vapor in the lowermost stratosphere in models has impacts in the troposphere, potentially shifting storm tracks and weather patterns, and causes uncertainties in simulating surface climate. The emergence of clear tropospheric impacts in our model experiment with constrained sea-surface temperatures ("Methods" section) is consistent with results from more idealized studies[3], and the surface temperature response in fully coupled atmosphere-ocean models can be expected to be even larger[32].

## Multi-model assessment

The presented model experiment shows a strong causal link between water vapor changes in the lowermost stratosphere and atmospheric circulation in both troposphere and stratosphere. This raises the question as to whether a similar mechanism may help to explain observed differences in recent climate model inter-comparisons. Indeed, a robust inter-model correlation is found among CCMI–2022 and CMIP6 models between water vapor mixing ratios and temperatures in the lowermost stratosphere (Fig. 4a), such that those models with higher water vapor mixing ratios also simulate lower temperatures in that region (correlation coefficients of −0.57 and −0.56 in the NH and SH, significant at 95% confidence level). This correlation maximizes for hemispheric fall, the season with highest water vapor values in the extratropical lowermost stratosphere.

A colder lowermost stratosphere implies a decreased meridional temperature gradient in the subtropics (Fig. 4b) which is, in turn, associated with a change in the vertical gradient in zonal wind via thermal wind balance ("Methods" section) such that the subtropical jets intensify at their upper flanks (Fig. 4c). Latitudinal cross-sections through the inter-model correlation between lowermost stratospheric water vapor and zonal wind velocity show positive correlations at the upper flanks of the subtropical jets and negative correlations below the jet core in both hemispheres (Fig. 4e, f, Methods). The clear correlation pattern indicates a strengthening of the jets at their upper flanks and weakening at their lower flanks and hence an effective upward jet shift with increasing lowermost stratospheric water vapor across models. Furthermore, correlations change from negative to positive across the jet axis in the latitudinal direction (Fig. 4e, f), except in the SH during summer. This pattern indicates that increased lowermost stratospheric water vapor is associated with a strengthening of the NH subtropical jet on its poleward side and a weakening on its equatorward side, resulting in a net poleward jet shift which reaches downward into the troposphere. Especially in view of the various competing factors which influence atmospheric temperatures, as well as subtropical jet strength and location, the presented inter-model correlations appear remarkably strong (Fig. 4). Furthermore, the upward jet shift in the models modifies the propagation and breaking of atmospheric waves[33] and leads to an intensified stratospheric Brewer-Dobson circulation, with a high correlation between tropical upwelling and lowermost stratospheric water vapor among models (correlation coefficients 0.82/0.84 in NH/SH, Fig. 4d).

## Model uncertainty

The relationships found in the full-blown CMIP6 and CCMI–2022 model simulations are highly consistent with the dynamical mechanism identified in the causal EMAC model experiment (for details on the models see "Methods" section). Hence, the atmospheric circulation differences among CMIP6 and CCMI–2022 models appear indeed related, at least partially, to differences in the amount of lowermost stratospheric water vapor. Consequentially, the quality of representing the atmospheric circulation in climate models is likely affected by a moist bias in the lowermost stratosphere, which is characteristic of these models. Notably, a similar moist bias is also present in current generation meteorological reanalyses[34], such that also the representation of the atmospheric circulation in these data sets is possibly affected by the flawed water vapor distribution in the lowermost stratosphere.

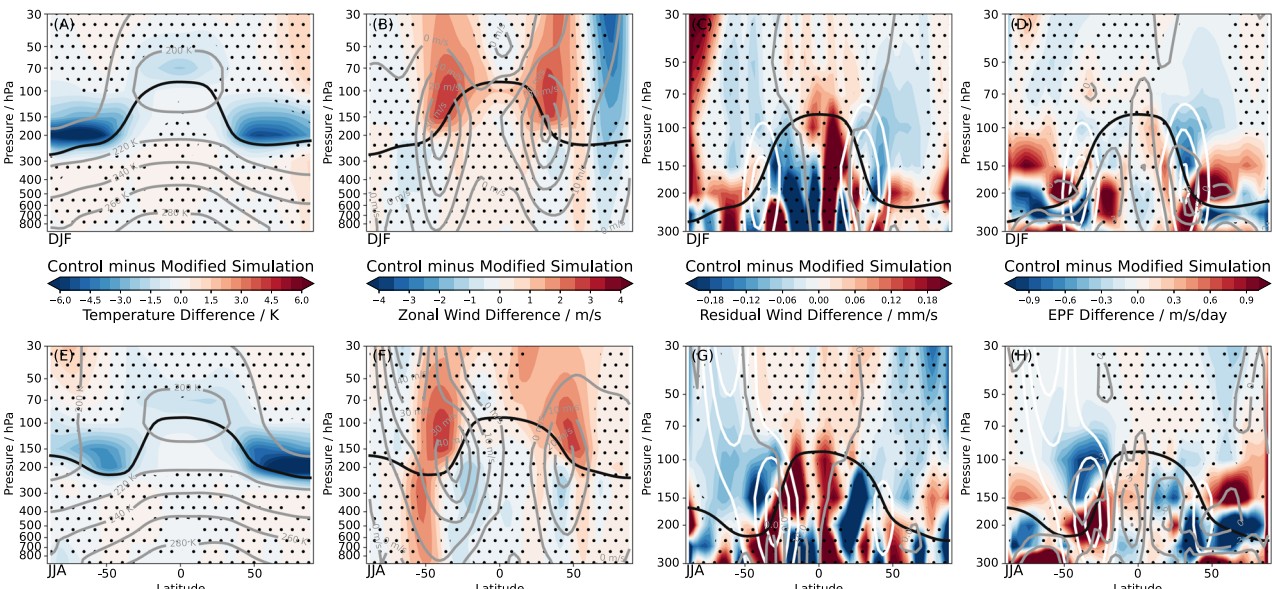

**Fig. 2 | Atmospheric circulation effects and dynamical mechanism induced by lowermost stratospheric water vapor changes. A, E** Temperature differences between control minus Lagrangian modified EMAC–CLaMS simulations are shown for boreal winter (December–February, top) and summer (June–August, bottom). Climatological temperatures from the control simulation are shown as gray contours, the WMO lapse rate tropopause as black line (for CMIP6 and CCMI2 calculated from ERA5 reanalysis). The same is shown for (**B, F**) zonal wind, (**C, G**) residual circulation vertical velocity $\overline{w}^*$, **D, H** resolved wave forcing quantified in terms of Eliassen-Palm flux divergence. For the latter variables, zonal wind contours from control simulation are shown as white lines. Dots indicate non-significant changes compared to year-to-year variability (at 95% confidence level).

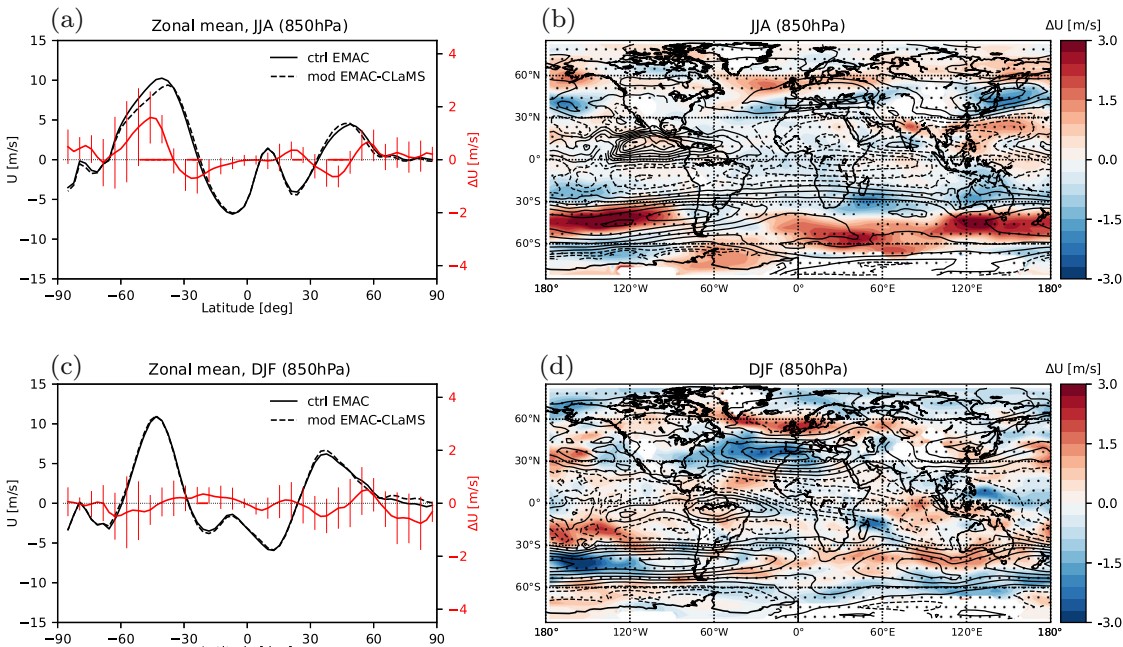

**Fig. 3 | Impact of stratospheric water vapor differences on near–surface circulation. a** Zonal mean zonal wind on 850 hPa pressure surface during boreal summer (June–August) from the control and the new Lagrangian modified EMAC–CLaMS simulation (black lines), with stratospheric water vapor in the Lagrangian simulation substantially reduced, and the difference "control" minus "modified" (red line). Error bars show 2-$\sigma$ standard error range related to year-to-year variability, and regions with changes being significant at 95% confidence level are marked with red coloring of the base line. **b** Difference map of zonal wind on 850 hPa between control and Lagrangian modified EMAC–CLaMS simulation for boreal summer. Black contours show the zonal wind reference climatology from control EMAC. Dotted regions indicate non-significant changes compared to year-to-year variability (at 95% confidence level). **c, d** Same, but for boreal winter season (December–February).

Recent studies have shown an important role of convectively lofted ice for the moisture budget of the lowermost stratosphere[35,36]. Thus, alternatively to the transport scheme, the model moist bias could be, at least partly, related to convectively lofted ice, and hence to the convection parameterization. However, additional model sensitivity tests show that advection is the only significant moistening process in the lowermost stratosphere in the EMAC model ("Methods" section), and therefore the moist bias in EMAC is very likely caused by advective transport. The role of convectively lofted ice in other models can not be

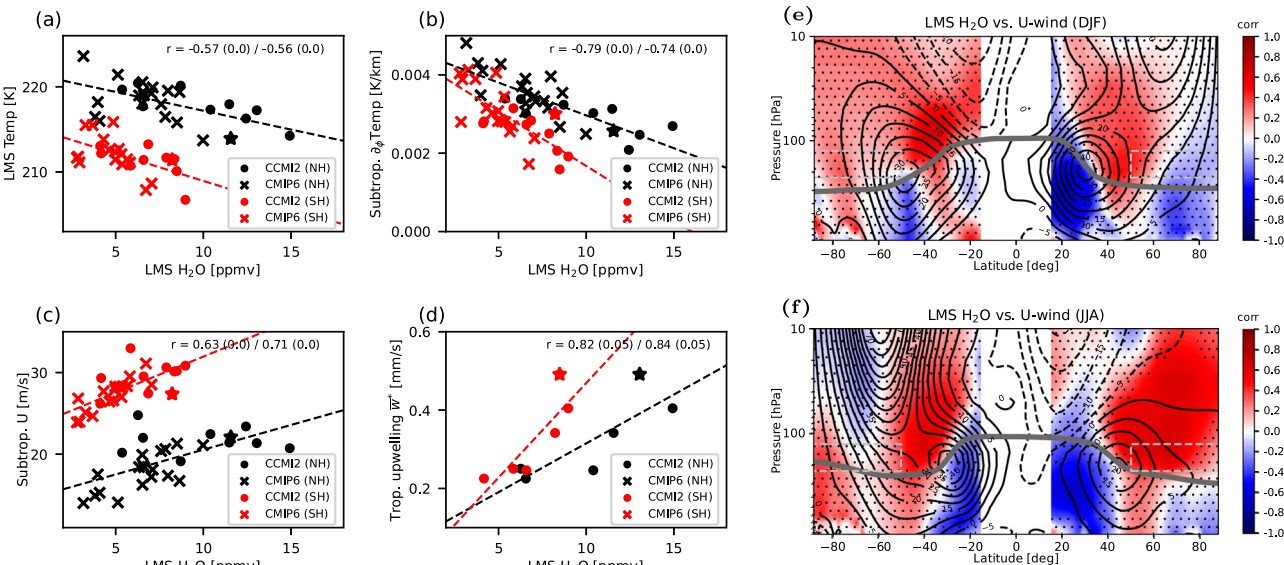

**Fig. 4 | Correlation of temperatures and circulation with stratospheric water vapor across climate models.** Shown are the inter-model correlations for CCMI−2022 and CMIP6 models, between the mean water vapor mixing ratio in the lowermost stratosphere (LMS, Northern hemisphere black, Southern hemisphere red) and **a** LMS temperature, **b** subtropical meridional temperature gradient, **c** subtropical jet intensity (zonal wind at upper jet flank), and **d** tropical upwelling (only available for a few models). For details on selected regions and available models see Methods. Data shown are from CCMI−2022 models (circles) and CMIP6 (crosses), with EMAC (in CCMI−2022) highlighted with stars. Pearson correlation coefficient values are given in each plot for Northern/Southern hemisphere data,

together with the associated p-values estimating statistical significance (in brackets, rounded to 2 decimals). **e** Pearson correlation coefficients between mean LMS water vapor mixing ratios and local zonal wind across CCMI−2022 and CMIP6 models at each pressure−latitude grid point for December−February, and **f** for June−August. Climatological zonal wind for the multi-model mean is shown as black contours, the WMO lapse rate tropopause (from ERA5 reanalysis) as a thick gray line, and the regions for calculating mean LMS water vapor are highlighted as gray dashed boxes. Correlation coefficients in the Northern hemisphere are calculated with Northern hemisphere LMS water vapor, and respectively for the Southern hemisphere. Dots indicate where correlations are not significant at 95% confidence level.

assessed here and could be a promising topic for future follow-up research.

The model-simulated response to future climate change involves a robust increase of stratospheric water vapor, strongest in the extratropical lowermost stratosphere[7,13], which induces a positive climate feedback[11,12]. Remarkably, the radiative and dynamical effects of stratospheric water vapor in the present climate due to differences in the transport scheme and between models, as found in this study, are of similar magnitude as the response to future warming[11]. This analogy corroborates the important role of stratospheric water vapor for atmospheric circulation and the robustness of involved dynamical mechanisms. However, a significant uncertainty in the simulated stratospheric water vapor feedback and related impacts on atmospheric circulation is likely related to the representation of water vapor transport processes in models, in particular in the lowermost stratosphere. Climate model projections of changes in stratospheric circulation are connected with changes in lower tropospheric winds and have a determining role in projections of societally relevant quantities like storminess and precipitation at the regional level[37]. An improved understanding of the causes of model biases for present-day circulation may lead to improved projections of climate change.

The presented model experiment and analyses show evidence that the water vapor content of the lowermost stratosphere exerts a first order effect on the global atmospheric circulation, ranging from the stratosphere to the troposphere and even affecting near-surface circulation patterns at 850 hPa. Commonly used model transport schemes may cause substantial biases in lowermost stratospheric water vapor, which can be alleviated with a newly incorporated, less diffusive Lagrangian transport scheme, resulting in far more realistic water vapor transport. Remarkably, the stratospheric cooling (up to 10 K) related to the stratospheric moist bias induces similar effects as

global warming, including an upward extension of subtropical jets, a poleward shift of the eddy-driven jet, and a strengthening shallow branch of the stratospheric circulation. Through this circulation pathway, lowermost stratospheric water vapor variations can affect tropospheric circulation and weather, and may cause strong regional climate effects.

Furthermore, the apparently large biases in simulated lowermost stratospheric water vapor in current climate models likely cause substantial biases not only in local temperatures but also in atmospheric circulation. Correcting the water vapor bias in the lowermost stratosphere in models has not come under focus in the recent past, but appears to be a fruitful topic for future climate research. The fact that lowermost stratospheric water vapor is affected by very different processes acting across scales, from large-scale stratospheric circulation to isentropic mixing, convection, ice microphysics and small-scale dynamics, makes related research especially challenging. In particular improving the representation of tracer transport in the lowermost stratosphere in models opens up a promising avenue to improve the reliability of future projections.

## Methods
### Water vapor observations
Water vapor observations used in this paper are taken from the publically available Stratospheric Water and OzOne Satellite Homogenized (SWOOSH) data set. SWOOSH comprises a merged satellite observation data set for stratospheric water vapor and ozone, based on measurements from the SAGE-II/III, UARS HALOE, UARS MLS, and Aura MLS instruments. The data set covers the time period from about 1984 to present. Here we used the monthly mean, zonal mean SWOOSH data on pressure levels. Further details about SWOOSH, e.g. regarding the number of observations, instrument uncertainty, the data homogenization method, or data access are described in the SWOOSH documentation paper[38].

## EMAC model experiment

The ECHAM/MESSy Atmospheric Chemistry model (EMAC) is a chemistry climate model based on ECHAM as dynamical core and coupled via the Modular Earth Submodel System (MESSy) to physical and chemical processes[23]. The EMAC simulations carried out for this study have been designed for middle atmosphere research and use a T42 spectral resolution, corresponding to a horizontal quadratic Gaussian grid of ~2.8 × 2.8 degrees horizontal resolution, and 90 vertical layers covering the atmosphere from the surface to 0.01 hPa. These simulations are free-running. Sea surface temperatures (SSTs) were prescribed with ERA-interim values. No interactive chemistry was simulated, but the radiatively active substances ($CO_2$, $O_3$, $N_2O$, CFC-11, and CFC-12, as well as tropospheric and stratospheric aerosol) have been prescribed from the ESCiMo simulation RC1-base-07[23]. Water vapor and methane, which causes a significant source of water vapor in the stratosphere by methane oxidation, have been calculated online.

In the "control" EMAC simulation, trace gas transport is calculated with the standard EMAC Flux-Form Semi-Lagrangian transport scheme. In the "modified" EMAC-CLaMS simulation, a newly incorporated Lagrangian transport scheme is applied for the calculation of water vapor transport in the stratosphere[26,27]. For both simulations, a ten year spin-up period was simulated using the control settings starting from January 1, 1970, initialized from ESCiMo simulation RC1-base-07. Thereafter, the Lagrangian modified simulation has been branched off and three more years have been simulated with either setting to let temperatures and circulation further adjust. After this period, a ten year production period was performed, using the control and the modified settings, and all analyses are based on this period. The simulations have been continued for 5 more years and all presented results are largely insensitive to varying the averaging period.

The Lagrangian transport scheme was developed within the Chemical Lagrangian Model of the Stratosphere (CLaMS)[25] and was optimized for transport calculations in stratospheric regions with steep tracer gradients. Due to its purely Lagrangian nature, CLaMS is almost free of numerical diffusion, except weak effects arising from the interpolations between parcel positions and the regular grids of meteorological fields and those needed for the EMAC radiation calculations. The advective transport in CLaMS is based on the calculation of 3D trajectories in a diabatic framework, hence using potential temperature as vertical coordinate in the stratosphere[39]. Unresolved atmospheric mixing due to small-scale turbulence is represented in CLaMS by a mixing parameterization with mixing strength depending on the shear rate in the large-scale flow via a critical Lyapunov exponent[25].

In past studies, CLaMS transport in reanalysis-driven offline simulations has been proven particularly advantageous for simulation of stratospheric water vapor[40]. In the "modified" EMAC-CLaMS simulation, the water vapor field below ~250 hPa (i.e. the closest model level to that pressure surface) is taken from the "control" EMAC simulation, and above the water vapor transport is calculated with the new Lagrangian transport scheme. This Lagrangian water vapor field is coupled to the EMAC radiation scheme. Hence, in the "modified" EMAC-CLaMS simulation a purely Lagrangian water vapor transport computation is realized throughout the stratosphere, which then affects atmospheric temperatures and circulation. To enable separation of the effects of stratospheric water vapor alone, ice and liquid water content as well as other cloud parameters from the control simulation are used in both simulations.

To investigate a potential contribution of convectively lofted ice due to the convection parameterization in "control" EMAC to the moisture difference compared to "modified" EMAC-CLaMS, the different water vapor tendencies have been diagnosed in a sensitivity simulation (Supp. Fig. 6). It was found that for the lowermost stratosphere zonal mean water vapor budget the only significant positive tendency (related to moistening) is due to advection. The potential

contribution due to convectively lofted, and subsequently sublimating, ice is included in the EMAC cloud tendency which is overall negative, meaning that cloud processes in that region decrease water vapor. To what extent these findings can be generalized to other models can not be answered here, and similar sensitivity studies disentangling the effects of different processes in the moisture budget of the lowermost stratosphere in models are highly recommended.

Comparison of relevant EMAC characteristics to other CCMI-2022 models in Fig. 4 (stars) shows that EMAC values are well within the range of other models. Also, past multi-model inter-comparisons show that the stratospheric circulation in EMAC compares well with other climate models[41], and that EMAC can be seen as representative for the current climate model suite. Nevertheless, a potential sensitivity of the results of the Lagrangian model experiment to the choice of base model can not be excluded unless similar experiments have been conducted also for other models.

The somewhat idealized model set-up used here, based on an atmospheric GCM and modifying only water vapor transport above 250 hPa, allows a clear separation of the radiation and circulation effects due to changes in lowermost stratospheric water vapor. It should be noted that the fixed SSTs suppress the slow climate feedbacks and that surface temperature changes will be even larger when including coupling to the ocean[32]. Hence, the presented near-surface impacts on circulation and temperature at 850 hPa can likely be interpreted as lower limits of the response to changes in lowermost stratospheric water vapor.

## Climate model intercomparison projects

The Chemistry-Climate Model Initiative (CCMI-2022) is a multi-model research activity conceived in support of the WMO/UNEP Scientific Assessment of Ozone Depletion Report 2022[42]. Here we use the refD1 historical hindcast simulations which cover 1960-2018 and have prescribed SSTs and SICs as boundary conditions following the HadISST1 datasets[43]. Concentrations of long-lived greenhouse gases follow the CMIP6 historical database up to 2014 and extended to the end of 2018 following the SSP2-4.5 scenario[44]. Ozone depleting substances follow the WMO baseline scenario. In order to ensure consistency between models, the Quasi-Biennial Oscillation (QBO) is nudged towards the observations. The 10 CCMI-2022 models with the necessary data to be used in the inter-model correlation analysis of this study (see below) are ACCESS-CM2-Chem, CCSRNIES-MIROC32, CESM2-WACCM, CMAM, EMAC, GEOSCCM, IPSL-CM6A-ATM-LR-REPROBUS, NIWA-UKCA2, SOCOL, UKESM1-StratTrop. The Transformed-Eulerian Mean (TEM) diagnostics, including the vertical component of the residual circulation and Eliassen-Palm Flux (EP-flux) divergence, have been computed following the guidelines of ref. 45.

he Coupled Model Intercomparison Project, Phase 6 (CMIP6) is a multi-model research activity conceived in support of the Sixth Assessment of the IPCC (AR6). The historical simulations cover the period from 1850 to 2014. These are fully coupled model simulations, and the external forcings, including solar variability, volcanic aerosols, and anthropogenic emissions of GHGs and aerosols, follow observations[46]. Models without ozone chemistry have prescribed time-varying ozone concentrations. We include the same 18 models as investigated in a recent stratospheric water vapor intercomparison[7] in the correlation analysis (AWI-ESM-1-1-LR, BCC-CSM2-MR, BCC-ESM1, CESM2, CESM2-FV2, CESM2-WACCM, CESM2-WACCM-FV2, CNRM-CM6-1, CNRM-ESM2-1, E3SM-1-1, GFDL-CM4, IPSL-CM6A-LR, MPI-ESM-1-2-HAM, MPI-ESM1-2-HR, MPI-ESM1-2-LR, MRI-ESM2-0, NorESM2-MM, UKESM1-0-LL), as these have been shown to simulate reasonable stratospheric water vapor.

## Inter-model correlations

Inter-model relations across CMIP6 climate and CCMI-2022 chemistry climate models from historical simulations are used to study the

effects of lowermost stratospheric water vapor on atmospheric circulation from a multi-model perspective. For that purpose, a mean lowermost stratospheric water vapor index was correlated with atmospheric temperatures, zonal winds and stratospheric residual circulation tropical upwelling across models. The specific regions to calculate climatological averages of the different variables for all models are: (i) for water vapor and atmospheric temperatures the lowermost stratosphere between 125–225 hPa and 50–90°N/S, (ii) for the subtropical meridional temperature gradient, the subtropics between 125-175 hPa and 20–50°N/S, (iii) for the subtropical jet intensity at the upper jet flank the subtropics between 125–225 hPa and 40–50°N, and (iv) for residual circulation tropical upwelling the tropical lower stratosphere between 60–80 hPa and 20°S–20°N. Only 6 CCMI–2022 models were available with residual circulation upwelling data (ACCESS-CM2-Chem, CCSRNIES-MIROC32, CMAM, EMAC, GEOSCCM, UKESM1–StratTrop), such that the respective correlation is based on a rather small ensemble. The correlations are quantified using the Pearson correlation coefficient, and the statistical significance, is assessed with the respective $p$-value (with a $p$-value of 0.05 implying statistical significance at 95% confidence level, etc.).

### Dynamical balances and subtropical jets

The radiative effect of stratospheric water vapor changes on atmospheric temperatures modifies long-wave cooling and hence induces decreasing local temperatures for increasing water vapor mixing ratios. Therefore, water vapor induced cooling in the lowermost stratosphere causes a decreased equator-to-pole temperature gradient. A change in meridional temperature gradient $\Delta(\partial_y T)$, in turn, is related to a change in the vertical zonal wind gradient $\Delta(\partial_p u)$ via the thermal wind relation

$$\Delta(\partial_z u) = -\frac{R}{Hf}\Delta(\partial_y T), \tag{1}$$

where $u$ is the zonal wind speed, $T$ atmospheric temperature, $f$ the Coriolis parameter, $R$ the gas constant, $H$ scale height, $z$ altitude, and $y$ the northward displacement. Hence, the lowermost stratosphere cooling due to water vapor increase induces a strengthening of the subtropical jet stream at its upper flank and upward jet shift.

### ERA5 reanalysis

ERA5 is the newest reanalysis from the European Centre for Medium-Range Weather Forecasts (ECMWF) and covers the period from 1949 to present[47]. The data production lags real time by about 2 months. ERA5 is based on 4D-Var data assimilation of the ECMWF Integrated Forecast System (in cycle CY41R2). The resolution of the data is about 30 km (T639) in the horizontal. The vertical range is from the surface to 0.01 hPa pressure level and comprises 137 hybrid levels. The ERA5 reanalysis data is available hourly. The ERA5 tropopause climatologies shown in this paper have been calculated from 6-h (00:00, 06:00, 12:00, and 18:00 UTC) ERA5 data with a truncated 1° × 1° reduced horizontal and full vertical resolution, as provided by the ECMWF MARS processing system.

## Data availability

CMIP6 model data is publically available at https://esgf-node.llnl.gov/search/cmip6. CCMI2 model data is available after requesting access, as clarified at https://blogs.reading.ac.uk/ccmi/ccmi-2022_archive. SWOOSH observation data is publically available at https://csl.noaa.gov/groups/csl8/swoosh. ERA5 data is publically available at https://cds.climate.copernicus.eu/cdsapp#!/dataset/reanalysis-era5-single-levels. The post-processing data used to produce the figures in this paper is available on request to the corresponding authors.

## Code availability

The EMAC and EMAC-CLaMS models, as well as the related configurations, are available in the Modular Earth Submodel System (MESSy) git database. Access to the MESSy codebase requires that the requestee be a member of a MESSy Consortium affiliation institution and that the requestee complete a license application. Detailed information is available at https://messy-interface.org/licence/application. The code used to produce the figures in this paper is available on request to the corresponding authors.

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

## Acknowledgements

We thank Nicole Thomas for programming support, and Paul Konopka, and Rolf Müller for helpful discussions. This research has been sup-ported by the Deutsche Forschungsgemeinschaft (DFG, German Research Foundation; TRR 301, project ID 428312742), and also by the Helmholtz Association (grant no. VH-NG-1128, Helmholtz Young Inves-tigators Group A–SPECi). We also gratefully acknowledge the comput-ing time for the EMAC simulations, which was granted on the supercomputer JUWELS at the Jülich Supercomputing Centre (JSC) under the VSR project ID CLAMS–ESM. SAS was supported by the Chemistry-Climate Modeling (CCM) work-package of the NASA Model-ing, Analysis, and Prediction Program. GEOSCCM CCMI simulations were performed at the NASA Center for Climate Simulation. CCSRNIES-MIROC32 simulations were performed using the NIES supercomputer. JK and NLA would like to thank the Met Office CSSP-China programme for funding received through the POzSUM project. UKESM1-StratTrop integrations were performed on MONSooN2 (a collaborative High-Performance Computing facility funded by the Met Office and the Nat-ural Environment Research Council) and data processing was performed on JASMIN, the UK collaborative data analysis facility. O.M. and G.Z. were supported by the NZ Government's Strategic Science Investment Fund (SSIF) through the NIWA programme CACV. The authors wish to acknowledge the use of New Zealand eScience Infrastructure (NeSI) high-performance computing facilities, consulting support and/or training services as part of this research. New Zealand's national facilities are provided by NeSI and funded jointly by NeSI's collaborator institu-tions and through the Ministry of Business, Innovation & Employment's Research Infrastructure programme (https://www.nesi.org.nz). ER and TE are grateful to the Swiss National Science Foundation for the SOCOL model support through the project POLE (grant 200020-182239). Cal-culations with the SOCOL model were performed under a grant from the Swiss National Supercomputing Centre (CSCS) under projects S-901 (ID no. 154), S-1029 (ID no. 249), and S-903. We also thank Karen Rosenlof and Sean Davis for creating the merged satellite SWOOSH database. We acknowledge the modeling groups for making their simulations avail-able for this analysis, the joint WCRP SPARC/IGAC Chemistry-Climate Model Initiative (CCMI) for organizing and coordinating the model data analysis activity, and the Centre for Environmental Data Analysis (CEDA)

for collecting and archiving the CCMI model output. Finally, we acknowledge the World Climate Research Programme, which, through its Working Group on Coupled Modelling, coordinated and promoted CMIP6. We thank the climate modeling groups for producing and making available their model output, the Earth System Grid Federation (ESGF) for archiving the data and providing access, and the multiple funding agencies who support CMIP6 and ESGF.

## Author contributions
E.C. and F.P. designed the MESSy model experiment with advice from P.J. and M.R.; E.C. performed the MESSy model development, executed the experiments, and processed the resulting data. R.B. provided tools to calculate some quantities used in the experimental data processing. E.C. and F.P. performed collection, processing, and analysis of data from all other sources. E.C. and F.P. produced the figures in the manuscript. M.A., N.L.A., H.A., S.B., F.D., P.J., J.K., D.K., O.M., D.P., E.R., S.S., G.Z., and T.E. provided data from and advice on the model intercomparison projects. E.C., F.P., and T.B. wrote the manuscript with comments and assistance from R.B., M.A., N.L.A., H.A., S.B., F.D., P.J., J.K., D.K., O.M., D.P., E.R., S.S., G.Z., T.E., and M.R.

## Funding

## Competing interests
The authors declare no competing interests.

## Additional information

[1]Institute for Energy and Climate Research: Stratosphere (IEK–7), Research Center Jülich, Jülich, Germany. [2]Institute for Atmospheric and Environmental Research, University of Wuppertal, Wuppertal, Germany. [3]Meteorological Institute Munich, Ludwig Maximilians University of Munich, Munich, Germany. [4]Earth Physics and Astrophysics Department, Universidad Complutense de Madrid, Madrid, Spain. [5]National Centre for Atmospheric Science (NCAS), University of Cambridge, Cambridge, UK. [6]Yusuf Hamied Department of Chemistry, University of Cambridge, Cambridge, UK. [7]National Institute for Environmental Studies, Tsukuba, Japan. [8]Laboratoire de Météorologie Dynamique (LMD/IPSL), Palaiseau, France. [9]Commonwealth Scientific and Industrial Research Organization (CSIRO) Environment, Aspendale, VIC 3195, Australia. [10]Institut für Physik der Atmosphäre, Deutsches Zentrum für Luft- und Raumfahrt (DLR), Oberpfaffenhofen, Germany. [11]Atmospheric Chemistry Observations and Modeling Laboratory, National Center for Atmospheric Research, Boulder, CO 80301, USA. [12]National Institute of Water and Atmospheric Research, Wellington, New Zealand. [13]Climate Research Branch, Environment and Climate Change Canada, Montreal, Canada. [14]Physikalisch-Meteorologisches Observatorium, Davos World Radiation Center, Davos Dorf, Switzerland. [15]Goddard Earth Sciences Technology and Research (GESTAR-II), Morgan State University, Baltimore, MD 21251, USA. [16]NASA Goddard Space Flight Center, Greenbelt, MD 20771, USA. [17]These authors contributed equally: Edward Charlesworth, Felix Plöger. ✉e-mail: e.charlesworth@fz-juelich.de

