## [Peer Review File · Nature Communications]

Stratospheric water vapor affecting atmospheric circulationREVIEWER COMMENTS

Reviewer #1 (Remarks to the Author):

This paper discusses the lower-stratospheric moist biases that still exist in climate models (relative to satellite observations) in a clear and concise manner. Using a model experiment that includes a lagrangian transport scheme of water vapor into the stratosphere, they find that the moist bias is likely caused by the transport scheme which has a large effect on atmospheric circulation. Therefore, they make highlight the important of how water vapor transport into the stratosphere is modeled and emphasize that a less-diffusive lagrangian transport scheme (similar to what they use) could help alleviate the moist bias.

Regarding the questions as to whether this article is appropriate to Nature Communications: I think the demonstration that a lagrangian transport scheme as described can help more realistically model the transport of water vapor into the stratosphere is noteworthy especially to those who are in model development. The author's claims are well supported, and the methods are well explained. Therefore, I only have one minor comment (listed below), but I believe this paper is ready and appropriate for publication.

Page 12 (in the Climate model intercomparison projects subsection of the methods) – you have “WMO/UNEP Scientific Assessment of Ozone Depletion Report” cited with (authors?)

Reviewer #2 (Remarks to the Author):

The paper reports interesting and impactful findings of an elegant study, where an alternative advection scheme is used to identify shortcomings in the long-standing issue in global climate models of simulating of extratropical lower stratospheric water vapour more accurately. Follow-on analysis connects the advection bias with impacts on global circulation. The methods are valid, however there are a couple of vague phrases that I think should be clarified. The reader would be helped by some revision to the figure colour scales. The results are significant because it appears most, if not all, global climate models would benefit from harnessing these results in future development.

I am signing this note because some of the comments are based on my very specific experience.
Jacob Willock Smith

Specific comments that require addressing:

1. para 4: There was a follow-up paper by Stenke (Stenke et al., 2009) that conducted the same study in a coupled chemistry-climate model. Why is that not referred to instead?

Figs 1,2 and supplementary figs 1, 3: Please specify somewhere which tropopause definition you are using throughout, and what dataset. Fig 4 states a lapse rate tropopause from ERA5, but it is not clear whether that is an exceptional case. The supplementary figures appear to show the same tropopause for all models, is it ERA5 or MMM tropopause? Also, Fig 4 caption: The ERA5 reanalysis dataset lacks citation. Either cite the ERA5 dataset, and describe in the methods, or replace the ERA5 data in the figure.

Fig 1. I think the representativeness of the global climate model underlying their advection scheme experiment (EMAC) should be addressed in the methods. From the figures, it appears tropopause level is relatively high (Fig 1 c-d vs. Fig 1 a-b), and the southern hemisphere winter jet may be stronger than MMM of CMIP6 and CCM1-2022 (Fig 1 g-h vs. Fig 1 e-f) which may influence the results. If CLAMS was coupled with one of the other ESMs/CCMs, would you still expect the significant changes to 850hPa winds? Please guide readers on this point.

4. First sentence: "down to the surface" is misleading. As far as I can see, all of you results are

<850hPa. This should be rephrased.

5. EMAC - para 1: "present climate state". Please be specific. Is it a particular year, or climatology, or boundary conditions? Does it align with any of the CMIP6 or CCM1-2022 configurations?

5. EMAC - para 2: When you say "branched off", how do you initialise your Lagrangian tracer distribution above 250hPa? Do you initialise at zero, or the 'control-EMAC' h2o field?

5. EMAC - para 3 and 3.2 end of para 1: Regarding water budgets, the text indicates you have checked the convective tendency of water vapour, which I find odd because water vapour is an indirect result of convection. Convection injects ice, not liquid water or water vapour, at the altitudes approaching the UTLS (Dessler et al. 2016, Smith et al. 2022), and in my experience GCMs calculate sublimation separately to convection, so the sensitivity of your results to tendencies of sublimation and convective ice need to be checked separately. Please check them, or make clear whether your analysis has checked the effect on water vapour arising from convection. Also, this may only affect the extratropical stratospheric overworld, but how do methane oxidation contributions compare?

5. CCM1 - para 1: "QBO nudged". I am concerned that this will have a strong effect on the stratospheric circulation patterns you have investigated, possibly overshadowing the results in Fig. 2. Can you comment?

Supplementary Figs 1-4: colourbar scales are not described. No colorbar is provided, and I can't see any labelling of h2o difference contours as the caption mentions. Please add labels or colorbars or confirm it is the same as in main text Fig 1.

Supplementary Figs 1 and 2: The CMIP6 multi-model mean is not shown. Are there other subfigures missing? Either revise the displayed subfigures or remove the mention of MMM in the caption.

Specific comments that do not require addressing but may be worth considering:

2.1 - para 1: The weak cold bias, how might that be affected by reducing the wet bias in the UTLS?

2.1 - para 1: "300%" is not demonstrated in Fig. 1, the colour scales only show up to 200%. Either adjust the text or the figure (perhaps a version with an alternative colour scale in the supplement?), or refer to Fig 1 before stating the 300% bias and not just after (the mention of a 400% bias later in 2.1 Para 3 is more reasonably laid out).

2.2 - para 1: "10K" Again, not shown directly in the image. From experience, I know colourbars are difficult on latitude-height figures, and you are trying to balance the display of general differences and maximum differences, but the maxima are what you point out in the text and they are not quantified in any of Figs 1, 2A and 2E. Perhaps consider alternative colour scales that show the maximum values referred to in the text.

4. Final sentence: I think this sentence could be strengthened, it is much weaker than earlier summary statements.

5. EMAC - para 2: Is 3 years enough time for the tracer, temperature and circulation to adjust? I ran a similar (although simpler) tracer transport experiment for my PhD thesis (focusing on the tropical stratosphere where transport times are longer) and it took many more years for the tracer concentrations to equilibrate, let alone temperature and circulation responses. Since you say you have run the model for a further 5 years and tested model climatologies up to 15 years, I think that is sufficient for the paper. But, I would be interested in seeing the results of this sensitivity analysis of adjustment and climatology times.

Technical corrections:

The author affiliations need revision.

1. para 1: "Arctic [a]mplification"?

Fig 1 caption: "as [a] thick black line"

Figs 3A and 3C and supplementary Fig 5: please revise legend labels to be more consistent with earlier figures, is it EMAC-FFSL and EMAC-CLAMS, or is it control and modified EMAC?

Fig 4 caption: "as [a] thick grey line"

5. IMC para 1: "[.] The correlations..."

5. CMIPs para 1: extraneous reference ("author?") to remove

5. DBSJ final line: "...and and..."

Supplementary Figure 5 caption is incomplete, requires a description of subfigure b.

References

Dessler, A. E., Ye, H., Wang, T., Schoeberl, M. R., Oman, L. D., Douglass, A. R., ... Portmann, R. W. (2016). Transport of ice into the stratosphere and the humidification of the stratosphere over the 21st century. *Geophysical Research Letters*, 43(5), 2323–2329. <https://doi.org/10.1002/2016GL067991>

Smith, J. W., Bushell, A. C., Butchart, N., Haynes, P. H., & Maycock, A. C. (2022). The Effect of Convective Injection of Ice on Stratospheric Water Vapor in a Changing Climate. *Geophysical Research Letters*, 49(9). <https://doi.org/10.1029/2021GL097386>

Stenke, A., Dameris, M., Grewe, V., & Garny, H. (2009). Implications of Lagrangian transport for simulations with a coupled chemistry-climate model. *Atmospheric Chemistry and Physics*, 9(15), 5489–5504. <https://doi.org/10.5194/acp-9-5489-2009>

Reviewer #3 (Remarks to the Author):

This paper presents new results demonstrating large biases in lowermost stratosphere water vapor (LMS H₂O) in current generation climate models, which influences global-scale temperatures, winds and circulations. The importance of H₂O is demonstrated by new model calculations incorporating Lagrangian transport, which produce a dryer LMS compared to standard simulations. The new calculations are in better agreement with satellite H₂O observations and highlight that standard models are too diffusive for LMS H₂O. Model comparisons show that the LMS H₂O biases are tied to temperature, wind and circulation biases that extend into the troposphere and are linked to key features such as strength of the tropical Brewer-Dobson circulation. I like how the simple model comparison studies can isolate these large effects linked to LMS H₂O, and the linkages are easy to understand via balanced dynamics. The idealized results are then backed up by analyses of CMIP6 and CCM2 climate models, which demonstrate the same behaviors and sensitivity to LMS H₂O amounts. This work highlights new understanding of the explicit role of LMS H₂O that has been suggested in previous work (e.g. stratospheric H₂O climate feedback studies), but is more explicit and clear in these results. The climate effects of biased LMS H₂O turn out to be as important as other order (1) processes in climate change simulations. Overall, I think that this is an excellent paper reporting new and exciting results that will be of substantial interest to the research community. It certainly points to the need for further research on sensitivities of climate models to detailed transport behavior. The paper is clearly written and the figures are appropriate. One minor comment is that Fig. 4c simply reflects thermal wind balance in the models (not surprising), and could maybe be replaced by some other diagnostic directly related to LMS H₂O, such as subtropical jet intensity.

Reply to Reviewer 1

We thank the Reviewer for the positive evaluation of the manuscript. In the following, we address the minor comment raised (Reviewer's comment in italics). Text changes in the manuscript are highlighted in color.

General comment:

This paper discusses the lower-stratospheric moist biases that still exist in climate models (relative to satellite observations) in a clear and concise manner. Using a model experiment that includes a lagrangian transport scheme of water vapor into the stratosphere, they find that the moist bias is likely caused by the transport scheme which has a large effect on atmospheric circulation. Therefore, they make highlight the important of how water vapor transport into the stratosphere is modeled and emphasize that a less-diffusive lagrangian transport scheme (similar to what they use) could help alleviate the moist bias.

Regarding the questions as to whether this article is appropriate to Nature Communications: I think the demonstration that a lagrangian transport scheme as described can help more realistically model the transport of water vapor into the stratosphere is noteworthy especially to those who are in model development. The author's claims are well supported, and the methods are well explained. Therefore, I only have one minor comment (listed below), but I believe this paper is ready and appropriate for publication.

Thanks for this very positive recommendation.

Minor comment:

Page 12 (in the Climate model intercomparison projects subsection of the methods) – you have “WMO/UNEP Scientific Assessment of Ozone Depletion Report” cited with (authors?)

The citation of the WMO report has been corrected and should be in line with Nature's citation guidelines now.

Reply to Reviewer 2

We thank the Reviewer Dr. Jacob Willock Smith for his insightful and detailed comments and the positive evaluation of the manuscript. In the following, we address all comments and questions raised (Reviewer's comments in italics). Text changes in the manuscript are highlighted in color.

General comment:

The paper reports interesting and impactful findings of an elegant study, where an alternative advection scheme is used to identify shortcomings in the long-standing issue in global climate models of simulating of extratropical lower stratospheric water vapour more accurately. Follow-on analysis connects the advection bias with impacts on global circulation. The methods are valid, however there are a couple of vague phrases that I think should be clarified. The reader would be helped by some revision to the figure colour scales. The results are significant because it appears most, if not all, global climate models would benefit from harnessing these results in future development.

Thanks for the overall positive recommendation, and especially for the very good and detailed comments. We think they are all very helpful for further improving the paper and address them thoroughly in the following.

Specific comments that require addressing:

1. para 4: *There was a follow-up paper by Stenke (Stenke et al., 2009) that conducted the same study in a coupled chemistry-climate model. Why is that not referred to instead?*

The cited paper (Stenke et al., 2008) was the first one conducting such a Lagrangian experiment, and their results regarding stratospheric water vapour and temperatures are closest related to our study. The follow-up paper focused more on other chemical species.

Figs 1,2 and supplementary figs 1, 3: Please specify somewhere which tropopause definition you are using throughout, and what dataset. Fig 4 states a lapse rate tropopause from ERA5, but it is not clear whether that is an exceptional case. The supplementary figures appear to show the same tropopause for all models, is it ERA5 or MMM tropopause? Also, Fig 4 caption: The ERA5 reanalysis dataset lacks citation. Either cite the ERA5 dataset, and describe in the methods, or replace the ERA5 data in the figure.

Thanks for pointing out these inconsistencies in notation! The tropopause shown is always calculated following the WMO definition. For EMAC and EMAC-CLaMS simulations we calculated the tropopause from the instantaneous model output. For SWOOSH satellite observations and CCMI2 and CMIP6 monthly means we calculated the tropopause from ERA5. This is now clearly stated in the revised manuscript, and also ERA5 reanalysis is described in the Methods section and Hersbach et al. (2019) is cited there.

Fig 1: I think the representativeness of the global climate model underlying their advection scheme experiment (EMAC) should be addressed in the methods. From the figures, it appears tropopause level is relatively high (Fig 1 c-d vs. Fig 1 a-b), and the southern hemisphere winter jet may be stronger than MMM of CMIP6 and CCMI-2022 (Fig 1 g-h vs. Fig 1 e-f) which may influence the results. If CLaMS was coupled with one of the other ESMs/CCMs, would you still expect the significant changes to 850hPa winds? Please guide readers on this point.

We agree that the generalization of the model experiment results to other climate models can strictly be proven only after the same experiment has been carried out also for other models. Realizing Lagrangian transport also in other models would be a future task for the different modelling groups which we deem highly beneficial for improving stratospheric transport and which we would highly recommend, and such recommendation is exactly one of the goals of the present paper.

However, the multi-model correlation figure (Fig. 4) already suggests that EMAC is not an outlier and the relevant circulation characteristics are well within the range of the other models. To illustrate this, we revised Fig. 4 and highlight the EMAC points as stars. In the Method section "EMAC model experiment" the representativity of EMAC is now discussed briefly with respect to the new Fig. 4, and referring to published multi-model comparisons:

"Comparison of relevant EMAC characteristics to other CCMI-2022 models in Fig. 4 (stars) shows that EMAC values are well within the range of other models. Also, past multi-model intercomparisons show that the stratospheric circulation in EMAC compares well with other climate models (Dietmueller et al., 2018), and that EMAC can be seen as representative for the current climate model suite. Nevertheless, a potential sensitivity of the results of the Lagrangian model experiment to the choice of base model can not be excluded unless similar

experiments have been conducted also for other models.”

4. First sentence: *”down to the surface” is misleading. As far as I can see, all of your results are 850hPa. This should be rephrased.*

We specified the wording and say now: *”... affecting near-surface circulation patterns at 850 hPa.”*

5. EMAC - para 1: *”present climate state”. Please be specific. Is it a particular year, or climatology, of boundary conditions? Does it align with any of the CMIP6 or CCM1-2022 configurations?*

The phrase “present climate state” refers to ERA-interim SST values from 1970 to 1990 (which is not the same data set used in the recent CMIP or CCM1 simulations), as described in the following two paragraphs.

5. EMAC - para 2: *When you say ”branched off”, how do you initialise your Lagrangian tracer distribution above 250hPa? Do you initialise at zero, or the ’control-EMAC’ h2o field?*

Both simulations, standard EMAC and Lagrangian EMAC–CLaMS, have been started in 1970 with initial conditions taken from existing EMAC ESCIMO simulations. For the first 10 spin-up years, the EMAC water vapour field has been coupled to the radiation. For the Lagrangian EMAC–CLaMS simulation, the Lagrangian water vapour field has been coupled to radiation from 1 January 1980 on. We clarified the model description “EMAC model experiment” in the Methods section.

5. EMAC - para 3 and 3.2 end of para 1: *Regarding water budgets, the text indicates you have checked the convective tendency of water vapour, which I find odd because water vapour is an indirect result of convection. Convection injects ice, not liquid water or water vapour, at the altitudes approaching the UTLS (Dessler et al. 2016, Smith et al. 2022), and in my experience GCMs calculate sublimation separately to convection, so the sensitivity of your results to tendencies of sublimation and convective ice need to be checked separately. Please check them, or make clear whether your analysis has checked the effect on water vapour arising from convection. Also, this may only affect the extratropical stratospheric overworld, but how do methane oxidation contributions compare?*

We fully agree with the Reviewer here that at altitudes of the lowermost stratosphere convection injects ice. Indeed, we had checked all model tendencies for water vapor including the tendency for cloud processes, which includes ice sublimation, but the description in the manuscript was unclear. As we find this remark particularly important, and the previous tendency check was only for a short simulation period, we have extended the EMAC sensitivity simulation with output of water vapor tendencies due to all simulated processes to cover a full year and added a new Fig. 6 to the supplement (also included in this reply, Fig. 1. This sensitivity study shows that in EMAC the only positive tendency in the lowermost stratosphere from about 200 hPa upwards (besides chemistry due to methane oxidation which indeed plays a role only higher in the stratosphere) is related to advection. Ice injection and sublimation is included in the “cloud” tendency, which is overall strongly negative. So the water vapor difference between the two EMAC model simulations is very likely related to the representation of advective transport in the model.

However, we can not rule out a significant ice moistening effect in other models (or in the real atmosphere). Hence, we clarified the respective text in the manuscript aiming for a more balanced discussion of these issues in the revised manuscript:

”Recent studies have shown an important role of convectively lofted ice for the moisture budget of the lowermost stratosphere (Dessler et al., 2016; Smith et al., 2022). Thus, alternatively to the transport scheme, the model moist bias could be, at least partly, related to convectively lofted ice, and hence to the convection parameterization. But additional model sensitivity tests show that advection is the only significant moistening process in the lowermost stratosphere in the EMAC model (Methods), and therefore the moist bias in EMAC is very likely caused by advective transport. The role of convectively lofted ice in other models can not be assessed here and could be a promising topic for future follow-up research.”

The related new part in the methods section is: *”To investigate a potential contribution of convectively lofted ice due to the convection parameterization in “control” EMAC to the moisture difference compared to “modified” EMAC–CLaMS, the different water vapor tendencies have been diagnosed in a sensitivity simulation (supplementary information). It was found that for the lowermost stratosphere zonal mean water vapor budget the only significant positive tendency (related to moistening) is due to advection. The potential contribution due to convectively lofted, and subsequently sublimating, ice is included in the EMAC cloud tendency which is overall negative, meaning that cloud processes in that region decrease water vapor. To what extent these findings can be generalized to other models can not be answered here, and similar sensitivity studies disentangling*

the effects of different processes in the moisture budget of the lowermost stratosphere in models are highly recommended.”

5. CCMI - para 1: *”QBO nudged”*. *I am concerned that this will have a strong effect on the stratospheric circulation patterns you have investigated, possibly overshadowing the results in Fig. 2. Can you comment?*

We agree with the Reviewer that nudging the QBO in climate models has a strong effect on atmospheric circulation. However, we think that considering simulations with nudged QBO here is advantageous for ensuring consistency in simulated QBO and focussing on UTLS circulation effects. Also, our analysis here is limited by the available simulations from CMIP6 and CCMI–2022 projects. Investigation of interactions between UTLS water vapour and QBO could be an interesting topic for future research.

Supplementary Figs 1-4: *colourbar scales are not described. No colorbar is provided, and I can't see any labelling of h2o difference contours as the caption mentions. Please add labels or colorbars or confirm it is the same as in main text Fig 1.*

A colorbar has now been included for each of the Supplementary Figures 1 through 4. We would also like to thank the Reviewer for the careful observation, as re-examination of these figures led us to notice that the model CNRM-MOCAGE (not included in our analysis) was erroneously included in Supplementary Figures 3 and 4. This model has been removed from those figures.

Supplementary Figs 1 and 2: *The CMIP6 multi-model mean is not shown. Are there other subfigures missing? Either revise the displayed subfigures or remove the mention of MMM in the caption.*

The CMIP6 multi-model mean is now included in Supplementary Figures 1 and 2.

Specific comments that do not require addressing but may be worth considering:

2.1 - para 1: *The weak cold bias, how might that be affected by reducing the wet bias in the UTLS?*

The remark on the cold bias here just states consistency between the weak dry bias in the stratospheric overworld in CMIP and CCMI models and a known tropopause cold bias, the remark does not refer to the UTLS wet bias. We can't rule out a weak influence from the UTLS wet bias on tropical tropopause temperatures (e.g., via the dynamical mechanism strengthening the stratospheric circulation, see Fig. 2), but the similarity between water vapour distributions in the overworld between the two EMAC experiment simulations in Fig. 1 shows that this effect can only be very small.

2.1 - para 1: *”300%” is not demonstrated in Fig. 1, the colour scales only show up to 200%. Either adjust the text or the figure (perhaps a version with an alternative colour scale in the supplement?), or refer to Fig 1 before stating the 300% bias and not just after (the mention of a 400% bias later in 2.1 Para 3 is more reasonably laid out).*

Thanks for pointing out this inconsistency in describing results in Fig. 1. The statement ”300%” is based on inspection of the plotted data and indeed can not be seen from the current figure. As the current colour bar shows the overall differences over the entire UTLS region best, we keep it and revised the text: ”...exceeding 200% for the multi-model mean (MMM) in the summer hemisphere...”.

2.2 - para 1: *”10K” Again, not shown directly in the image. From experience, I know colourbars are difficult on latitude-height figures, and you are trying to balance the display of general differences and maximum differences, but the maxima are what you point out in the text and they are not quantified in any of Figs 1, 2A and 2E. Perhaps consider alternative colour scales that show the maximum values referred to in the text.*

Similar to the above comment, the statement ”10 K” was based on inspection of the plotted data. We revised the text to be consistent with the color bar in Fig. 2: ”These temperature differences are well above 6 K around the 200 hPa level in the summer hemisphere (reaching peak values of about 10K, not shown).”

4. Final sentence: *I think this sentence could be strengthened, it is much weaker than earlier summary statements.*

Thanks for this suggestion! We strengthened the final sentence to: ”In particular improving the representation of tracer transport in the lowermost stratosphere in models opens up a promising avenue to improve the relia-

bility of future projections.”

5. EMAC - para 2: *Is 3 years enough time for the tracer, temperature and circulation to adjust? I ran a similar (although simpler) tracer transport experiment for my PhD thesis (focusing on the tropical stratosphere where transport times are longer) and it took many more years for the tracer concentrations to equilibrate, let alone temperature and circulation responses. Since you say you have run the model for a further 5 years and tested model climatologies up to 15 years, I think that is sufficient for the paper. But, I would be interested in seeing the results of this sensitivity analysis of adjustment and climatology times.*

Indeed, the necessary spin-up period for model experiments is a critical issue. In our case a 3 year spin-up period after starting the coupling of Lagrangian water vapour to radiation is enough, as we start with an already equilibrated simulation (10 years spin-up before without enabling radiative coupling of Lagrangian water vapour, see reply to specific comment above). Below is a figure which depicts time series for water vapour, temperature and zonal wind from the different model simulations, showing that all these variables for the Lagrangian simulation have been adjusted already after a few months after starting radiative coupling of Lagrangian water vapour (Fig. 2).

Nevertheless, we continued the simulations in the mean time for another 15 years (providing 30 years of Lagrangian simulation with radiative coupling enabled), and confirmed that all results are robust also regarding even longer periods for calculating climatologies.

Technical corrections:

The author affiliations need revision.

Thanks for checking that - has been revised!

1. para 1: *”Arctic [a]mplification”?*

Corrected.

Fig 1 caption: *”as [a] thick black line”*

Corrected.

Figs 3A and 3C and supplementary Fig 5: *please revise legend labels to be more consistent with earlier figures, is it EMAC-FFSL and EMAC-CLAMS, or is it control and modified EMAC?*

Thanks for detecting these inconsistencies in labelling. We corrected all figure labels and checked the text carefully that the labels ”ctrl EMAC” and ”mod EMAC–CLaMS” are used consistently throughout the revised version and the supplement.

Fig 4 caption: *”as [a] thick grey line”*

Corrected.

5. IMC para 1: *”[.] The correlations...”*

Corrected.

5. CMIPs para 1: *extraneous reference (”author?”) to remove*

Extraneous reference (WMO report) removed and reference corrected.

5. DBSJ final line: *”...and and...”*

Corrected.

Supplementary Figure 5: *caption is incomplete, requires a description of subfigure b.*

Figure 1: **Tendency contributions to lowermost stratospheric water vapor.** Tendencies for control EMAC simulated water vapour at (a) 215 hPa and (b) 100 hPa averaged over the middle latitude region (50° - 70° N). Shown is the seasonal cycle of tendencies for advection, cloud processes (including e.g. dehydration, evaporation, ice sublimation), convection, parameterized vertical diffusion, and the sum of all these individual contributions. At 100 hPa, the tendencies due to convection, clouds and vertical diffusion overlie the zero line.

A description of subfigure b has been added to the caption.

Figure 2: Time series of water vapour (top), temperature (second), zonal wind (third to fifth row) from the different EMAC simulations, at the UTLS locations indicated in the respective plots.

Reply to Reviewer 3

We thank the Reviewer for the positive evaluation of the manuscript and the good comment. In the following, we address the minor comment raised (Reviewer's comment in italics). Text changes in the manuscript are highlighted in color.

General comment:

This paper presents new results demonstrating large biases in lowermost stratosphere water vapor (LMS H₂O) in current generation climate models, which influences global-scale temperatures, winds and circulations. The importance of H₂O is demonstrated by new model calculations incorporating Lagrangian transport, which produce a dryer LMS compared to standard simulations. The new calculations are in better agreement with satellite H₂O observations and highlight that standard models are too diffusive for LMS H₂O. Model comparisons show that the LMS H₂O biases are tied to temperature, wind and circulation biases that extend into the troposphere and are linked to key features such as strength of the tropical Brewer-Dobson circulation. I like how the simple model comparison studies can isolate these large effects linked to LMS H₂O, and the linkages are easy to understand via balanced dynamics. The idealized results are then backed up by analyses of CMIP6 and CCM12 climate models, which demonstrate the same behaviors and sensitivity to LMS H₂O amounts. This work highlights new understanding of the explicit role of LMS H₂O that has been suggested in previous work (e.g. stratospheric H₂O climate feedback studies), but is more explicit and clear in these results. The climate effects of biased LMS H₂O turn out to be as important as other order (1) processes in climate change simulations. Overall, I think that this is an excellent paper reporting new and exciting results that will be of substantial interest to the research community. It certainly points to the need for further research on sensitivities of climate models to detailed transport behavior. The paper is clearly written and the figures are appropriate.

Thanks for this very positive recommendation.

Minor comment:

One minor comment is that Fig. 4c simply reflects thermal wind balance in the models (not surprising), and could maybe be replaced by some other diagnostic directly related to LMS H₂O, such as subtropical jet intensity.

Thanks for this suggestions! We agree that the thermal wind balance holds so well for the inter-model differences that showing the correlation between jet intensity and LMS water vapour would be more helpful. We therefore revised Fig. 4c to show the correlation between subtropical zonal mean zonal wind at the upper flanks of the subtropical jets (40-50°, 175 – 100 hPa) and LMS water vapor.

The new correlation between jet intensity and LMS water vapor is also very clear, correlation coefficients of 0.63 and 0.71 in NH and SH, respectively (both significant at 95% confidence level). Hence, the general argumentation in the paper remains as before, with the respective text slightly adjusted (second paragraph in Sect. 3.1: "A colder lowermost stratosphere implies a decreased meridional temperature gradient in the subtropics (Fig. 4b) which is, in turn, associated with a change in the vertical gradient in zonal wind via thermal wind balance (Methods) such that the subtropical jets intensify at their upper flanks (Fig. 4c).").

REVIEWERS' COMMENTS

Reviewer #2 (Remarks to the Author):

I thank the authors for considering the reviewer comments thoroughly, including the non-essential questions.

The paper is fit for publication but for one technical correction that is important to resolve: The revised supplementary figures 1 and 2 (that are meant to show CMIP6) have been mixed up with supp. figures 3 and 4 (CCMI2).

Reviewer #3 (Remarks to the Author):

The authors have done a thoughtful job of responding to the earlier reviewer comments and made appropriate changes. I recommend publication in present form.